# Isolation and Cultivation of a New Isolate of BTV-25 and Presumptive Evidence for a Potential Persistent Infection in Healthy Goats

**DOI:** 10.3390/v12090983

**Published:** 2020-09-04

**Authors:** Christina Ries, Ursula Domes, Britta Janowetz, Jens Böttcher, Katinka Burkhardt, Thomas Miller, Martin Beer, Bernd Hoffmann

**Affiliations:** 1Institute of Diagnostic Virology, Friedrich-Loeffler-Institut, Südufer 10, 17943 Greifswald-Insel Riems, Germany; christina.ries@fli.de (C.R.); Martin.Beer@fli.de (M.B.); 2Bavarian Animal Health Service, Senator-Gerauer-Straße 23, 85586 Poing, Germany; Ursula.Domes@tgd-bayern.de (U.D.); Britta.Janowetz@tgd-bayern.de (B.J.); Jens.Boettcher@tgd-bayern.de (J.B.); 3Aulendorf State Veterinary Diagnostic Centre, Löwenbreitestraße 18/20, 88326 Aulendorf, Germany; katinka.burkhardt@stuaau.bwl.de (K.B.); Thomas.Miller@stuaau.bwl.de (T.M.)

**Keywords:** Bluetongue virus, BTV, atypical BTV, serotype 25, persistent infection, goats

## Abstract

Recently, several so-called “atypical” Bluetongue virus (BTV) serotypes were discovered, including BTV-25 (Toggenburg virus), in Switzerland. Most “atypical” BTV were identified in small ruminants without clinical signs. In 2018, two goats from a holding in Germany tested positive for BTV-25 genome by RT-qPCR prior to export. After experimental inoculation of the two goats with the BTV-25 positive field blood samples for generation of reference materials, viremia could be observed in one animal. For the first time, the BTV-25-related virus was isolated in cell culture from EDTA-blood and the full genome of isolate “BTV-25-GER2018” could be generated. BTV-25-GER2018 was only incompletely neutralized by ELISA-positive sera. We could monitor the BTV-25 occurrence in the respective affected goat flock of approximately 120 goats over several years. EDTA blood samples were screened with RT-qPCR using a newly developed BTV-25 specific assay. For serological surveillance, serum samples were screened using a commercial cELISA. BTV-25-GER2018 was detected over 4.5 years in the goat flock with intermittent PCR-positivity in some animals, and with or without concomitantly detected antibodies since 2015. We could demonstrate the viral persistence of BTV-25-GER2018 in goats for up to 4.5 years, and the first BTV-25 isolate is now available for further characterization.

## 1. Introduction

Bluetongue virus (BTV) is a double stranded and segmented RNA virus within the family *Reoviridae,* genus *Orbivirus,* that causes bluetongue disease in ruminants [1]. The first approaches of serotyping BTV strains according to the neutralization capabilities of strain-specific sera were made in the 1960s in South Africa [2]. Since then, the virus neutralization test (VNT) has become the gold standard for serotype identification, and up to now 24 classical BTV serotypes are known (Mertens et al. 2004; OIE terrestrial manual). Nevertheless, with the rapid progress in genomics in recent decades, more and more BTV sequence data have become available, and the idea of typing BTV according to its genotype arose. In 2011, a working group suggested levels of maximum and minimum nucleotide (nt) and amino acid (aa) identities in segment-2 of the BTV genome as an alternative to the traditional typing methods [3]. A remarkably increasing number of novel serotypes have been described since the discovery of BTV-25 (Toggenburg Virus, TOV) in 2008 [4]. This group of newly discovered BTV-strains differs in several viral characteristics, but also at the molecular level from the classical BTV serotypes 1–24. Consequently, non-classical BTV serotypes are referred to as the group of “atypical” BTVs, distinct from the classical and notifiable BTV serotypes 1–24 [5,6]. Nevertheless, the OIE recommended the Pan-BTV-segment 10 RT-qPCR [7,8] in order to detect all BTV serotypes, including the known atypical BTVs. Recently, we established the Pan-BTV-Classic-S1-RT-qPCR assay, targeting BTV segment 1 for distinction between classical and atypical serotypes [9]. The discovery of TOV was followed by the description of BTV-26 in samples from symptomatic sheep in Kuwait [3]. In addition, BTV-26 antibody circulation was discovered in cattle and dromedaries in the Islamic Republic of Mauritania [10]. Interestingly, horizontal contact transmission could be demonstrated for BTV-26 [11,12], which is in sharp contrast to the insect vector dependent transmission dynamics of classical BTV serotypes. Furthermore, three variants of BTV-27 were detected in asymptomatic goats on Corsica [13]. The two putative novel serotypes, BTV XJ1407 from China [14] and BTV-X ITL2015 from Italy [6], were serologically and molecularly characterized, but still require assignment to a new serotype. For another BTV strain—isolated from a contaminated sheep pox vaccine in Israel—full-length sequence data are available, and an experimental infection of sheep was conducted [15,16]. The most recent BTV-strain description was the Tunisian BTV-Y TUN2017 strain in sheep [17]. 

The initially described BTV-25 (Toggenburg Virus—TOV) was detected in two different asymptomatic goat flocks in the Toggenburg region in Switzerland [4]. Similarly, to naturally infected goats, experimentally TOV-infected goats did not develop clinical signs typical for BTV, even though they exhibited a high virus replication rate [18]. Experimentally TOV-infected sheep also presented a very mild clinical disease consisting of minor BTV characteristic symptoms [18]. Horizontal transmission of TOV seems unlikely, as contact control animals did not get infected, and all swabs as well as milk and saliva samples revealed negative results [19]. It should be also mentioned that the systemic spread of TOV in infected goats was described as being rather slow [19]. Nevertheless, the high seroprevalence rate of naturally infected goat flocks in combination with an extremely low vector activity in Switzerland provided some indication for the presence of an efficient alternative transmission route [18,19]. Furthermore, there are indicators for transplacental infection, but additional studies were suggested for confirmation [18,20]. TOV RNA could be detected for up to 25 months and the infectivity of blood during that period was demonstrated [21]. The antibody response of experimentally infected animals was described as slow and weak [19]. All attempts of cell culture-based virus isolation of TOV remained unsuccessful [19]. Thus, for the use in virus neutralization tests (VNTs), a chimeric classical BTV/TOV virus was generated by reverse genetics [20]. In 2018, another TOV-related BTV strain (BTV-Z ITA2017) was described in the Piedmont region in Italy, and it could also not be cultivated [22]. This TOV-related strain was found in healthy goats and showed a high identity with TOV, both on the nucleotide (nt) and the amino acid (aa) level. Nevertheless, the serotype remained undefined, due to the failure of cELISA-positive sera to neutralize the reference and the atypical (including chimeric strains) BTV serotypes by serum neutralization (SN) [22]. For BTV-25 detection, two specific real-time RT-qPCR systems targeting segment 2 have been developed over the years [23,24].

Concerning the BTV situation in Germany, BTV-8 played a major role and was present from 2006 to 2009. Eradication was successful with the application of an obligatory BTV-8 vaccination program [25]. In February 2012, Germany was declared officially free of BTV until the re-emergence of BTV-8 in December 2018 [26]. In our study, the novel BTV-25 related virus (BTV-25-GER2018) detected in healthy goats in the southern part of Germany is further characterized. Full genomes were generated and phylogenetically analyzed, and for the first time a BTV-25-related virus could be propagated in cell culture. For consistent virus detection, a BTV-25-specific RT-qPCR assay targeting segment 2 and adapted to the new BTV-25-GER2018 strain was developed. Furthermore, the infected goat flock was monitored over a 14-month period with a series of blood and retrospective serum samples.

## 2. Materials and Methods 

### 2.1. RNA Extraction and RT-qPCR

Viral RNA of all EDTA blood samples was extracted either manually using the QIAamp Viral RNA Mini kit (Qiagen, Hilden, Germany) or the NucleoMagVET kit (Macherey-Nagel, Düren, Germany) with the help of a half-automated KingFisher platform (King-Fisher Flex magnetic particle processor, Thermo Fisher Scientific, Waltham, MA, USA). The RNA was amplified using the Pan-BTV-S10-RT-qPCR recommended by the OIE [7] and was considered positive when quantification cycle (Cq) values were <40. For initial serotype identification, two published available RT-qPCRs targeting segment 2 were used [23,24]. For further screening, an RT-qPCR (BTV-25-Mix13 assay) was developed based on all available sequence information of BTV-25-related strains. The forward primer BTV-25-2434-F (5′-GGT TCR ATT TGT TAT CGC TAC TAT A-3′) and the reverse primer BTV-25-2609-R (5′-ACA AGR CAC TTC TCT GGA TGT G-3′) were used in a 20 μM concentration, whereas the probe BTV-25-2494FAM (6-FAM-CCG GTT ATC ACT ACA AAG TTG GAC AC-BHQ1) was used in a 5 μM concentration for preparation of the primer–probe mixture. For process control, a heterologous control system was implemented and co-amplified in all PCR runs using the HEX channel [27]. The final composition of the RT-qPCR reactions was 1.25 μL of RNase-free water, 6.25 μL of 2× RT-PCR buffer, 0.5 μL of RT-PCR Enzyme Mix, 1 μL of primer-probe-mix-FAM, 1 μL of EGFP-mix1-HEX and 2.5 µL of the heat denatured template RNA. All RT-qPCRs were run on the CFX 96 real-time PCR cycler (Bio-Rad, Hercules, CA, USA) with the AgPath-ID™ One-Step RT-PCR Reagents of Applied Biosystems™ (Waltham, MA, USA). The temperature profile used was 10 min at 45 °C (reverse transcription), 10 min at 95 °C (inactivation of the reverse transcriptase/activation Taq polymerase) followed by 42 cycles of 15 s at 95 °C (denaturation), 20 s at 56 °C (annealing), and 30 s at 72 °C (elongation). Fluorescence values (FAM, HEX) were collected during the annealing step. The specificity of the BTV-25-Mix13 assay was tested in silico by BLAST search (https://blast.ncbi.nlm.nih.gov) and in vitro using available viral RNAs of all 24 classical BTV serotypes and further atypical BTV serotypes (BTV-26, three variants of BTV-27 and BTV-28). At all 5 blood sampling time points, EDTA blood was analyzed with the BTV-25 specific RT-qPCR (BTV-25-Mix13 assay). Furthermore, individual EDTA blood samples were tested in the BlueTYPE array as described previously [9].

### 2.2. Sequence Analysis

The sequences of the ten segments of BTV-25-GER2018 were generated using the HTS-SISPA technology [28] on the Illumina platform. In the first step, the viral RNA was extracted from the BTV-25 infected cell culture material using the MasterPure Complete DNA and RNA Purification Kit (Biozym Scientific GmbH, Hessisch Oldendorf, Germany). The cDNA first strand synthesis using the SISPA K8N random primer was performed by the qScript Flex cDNA synthesis Kit (Quanta Biosciences, Beverly, MA, USA). The heat denatured and immediately cooled down extracted RNA (10 µL) mixed with 0.5 µL of the 100 µM concentrated SISPA-K8N primer served as template. After including the 9.5 µL master mix preparation, reverse transcription was run with a total reaction volume of 20 µL using the temperature profile of 10 min 25 °C, 45 min at 42 °C, 5 min at 85 °C and cooling of 10 °C. For the second strand synthesis, the Second Strand cDNA Synthesis Kit-dNTP based (Applied Biological Materials Inc. (abm), Richmond, Canada) was used according to the supplier’s instructions. Briefly, the 20 µL cDNA template was heat denatured for 3 min at 95 °C and cooled down on ice for 5 min. Together with the 30 µL mastermix preparation the total amount of 50 µL reaction mix was incubated for 2.5 h at 16 °C. The double stranded cDNA was amplified using the K primer and the Phusion High Fidelity PCR Polymerase (New England Bio labs, Ipswich, USA). Therefore, 5 µL of purified double stranded (ds) cDNA was used as template in a 50 µL total reaction mix. The temperature profile used was 30 s at 98 °C followed by 35 cycles of 10 s at 98°C, 30 s at 60 °C and 30 s at 72 °C and in the end 5 min at 72 °C before permanent cooling at 10 °C. The generated double stranded cDNA was purified before and after the Phusion PCR with the sparQ PureMag Beads Kit (Quanta Biosciences, Beverly, MA, USA) by adding the 50 µL of double stranded cDNA to 40 µL of beads. After the procedure and according to the manufacturer’s instructions, the beads were re-suspended in 35 µL of 10 mM TRIS-HCL (pH 8.0). The amplified and purified ds cDNA was sent to Eurofins Genomics (Ebersberg, Germany) for sequencing on an Illumina platform. Raw data as fastq files were trimmed and assembled by mapping to the BTV-25 TOV reference sequences with the following accession numbers: GQ982522 (Seg-1), EU839840 (Seg-2), GQ982523 (Seg-3), GQ982524 (Seg-4), EU839841 (Seg-5), EU839842 (Seg-6), EU839843 (Seg-7), EU839844 (Seg-8), EU839845 (Seg-9), EU839846 (Seg-10) using the Geneious software v2019.2.3 (Biomatters Ltd., Auckland, New Zealand). For phylogenetic analyses, a multiple alignment of BTV sequences was performed by using the MAFFT alignment feature in the Geneious software. We included the identical BTV strain selection, representing known BTV serotypes as used in the publication of BTV-X-ITL2015 [6]. Phylogenetic trees of each of the 10 segments were created with MegaX [29] using the genetic distinction model Tamura–Nei and tree-built method UPGMA and for modification of the layout of the segments 2 and 6 FigTree v1.4.4 [30]. To assess the robustness of individual nodes on the phylogenetic trees, we performed a bootstrap analysis with 1000 replications. Furthermore, the consensus sequences of each of the ten segments of BTV-25-GER2018 were blasted against the nt/aa database of the NCBI for identifying the nearest molecular neighbors. The BTV-25-GER-2018 sequences obtained in this study were submitted to NCBI with the following accession numbers: LR798441 (Seg-1), LR798442 (Seg-2), LR798443 (Seg-3), LR798444 (Seg-4), LR798445 (Seg-5), LR798446 (Seg-6), LR798447 (Seg-7), LR798448 (Seg-8), LR798449 (Seg-9) and LR798450 (Seg-10).

### 2.3. Goat Flock in Bavaria

Two individual goats from a holding in the southern part of Bavaria (Germany) were initially tested for the presence of BTV genomes prior to export and tested BTV RNA positive. Subsequently, this goat flock was monitored and bled five times from August 2018 to October 2019. EDTA blood and serum samples were taken from the goats present at the time point of sampling. Two German breeds (“White German Edelziege” and “Colourful German Edelziege”) were present in the goat flock originating from Switzerland (*n* = 31), Baden-Württemberg (*n* = 51), Bavaria (*n* = 4) and their offspring (*n* = 52), respectively. The flock composition changed slightly over the different bleeding time points. In total, 23 goats were removed and another 20 were newly introduced over the monitoring period (July 2018—October 2019). The overview of the goat flock is given in Table 1. Furthermore, individual retrospective serum samples from February 2015, stored at −20°C, were available for investigation. 

### 2.4. Experimental Inoculation of Goats 

Two male, 6-month-old Thuringian goats were kept in the vector-free high containment buildings of the FLI, Isle of Riems, for diagnostic inoculation. The two goats (#19, #20) were inoculated with two EDTA blood samples from naturally BTV-25-GER2018 infected goats. Goat #19 was inoculated with the first (BH66/18_1; Cq-value = 26.1), and goat #20 with the second EDTA blood sample (BH66/18_2; Cq-value = 33.6). Then, 700 µL PBS-washed blood and 500 µL of unwashed blood were injected subcutaneously at two different injection sites. Both goats were monitored daily for clinical symptoms. EDTA blood and serum were taken regularly throughout the whole experiment. Goat #19 was kept until 31 dpi and goat #20 until 46 dpi. The respective experimental protocols were reviewed by the state ethics commission and approved by the competent authority (State Office for Agriculture, Food Safety and Fisheries of Mecklenburg-Vorpommern, Rostock, Germany; Ref. LALLF M-V/TSD/7221.3-1-048/19 from 07.11.2019).

### 2.5. Production of Antisera in Rabbits

Two rabbits were immunized with binary ethyleneimine (BEI)-inactivated BTV-25 full-virus cell culture material. Binary ethylenimine (BEI) was prepared freshly by cyclization of 0.1 M 2-bromoethylamine hydrobromide in 200 mM sodium hydroxide (NaOH) solution at 37 °C for 60 min [31]. Before inactivation, the two different BTV-25 GER2018 virus preparations had a titer of 10^3^ and 10^3.75^ TCID_50_/_mL_. Next, 2.7 mL of virus preparation was mixed with 0.3 mL of 0.1 M BEI and transferred into a new falcon after overnight incubation at 28 °C. After another incubation for 24 h at 28 °C the reaction was stopped by adding 0.3 mL of 200 mM sodium thiosulfate solution. The antigen preparation was aliquoted and stored at −70 °C until usage. The success of the inactivation procedure was confirmed by decreasing RT-qPCR Cq values of the cell culture material during 3 serial cell culture passages. Rabbits were inoculated subcutaneously three times at two-week intervals with 1 mL BTV-25-GER2018 inactivated antigen mixed with 100 µL of Polygen as adjuvant (MVP Adjuvants, USA). The final serum was collected at 56 dpv. The respective experimental protocols were reviewed by the state ethics commission and approved by the competent authority (State Office for Agriculture, Food Safety and Fisheries of Mecklenburg-Vorpommern, Rostock, Germany; Ref. LALLF M-V/TSD/7221.3–2-042/17).

### 2.6. Isolation of BTV-25-GER2018

Blood samples from naturally and experimentally infected goats were processed identically for the virus isolation experiments: 500 µL of EDTA blood was centrifuged (8000 rpm) for 2 min and the red blood cells were washed twice in 1 mL PBS and finally diluted in 500 µL PBS prior to lysis by 20 s ultrasound treatment at 30W (Sonifier 450, Branson Ultrasonics, USA). Additionally, unwashed blood of the experimentally infected goat was lysed by ultrasound treatment and used for virus isolation experiments as well. BHK-21 (BSR/5) cells (FLI cell culture collection number RIE0194) in T25 cm² cell flasks were incubated initially for three hours at 37 °C using the cultivation medium MEM with essential amino acids (FLI intern medium number ZB5d) supplemented with 10% FCS (fetal calf serum). Afterwards, the cells were inoculated with either 200 µL of washed blood cells from the BTV-25 positive non-experimental blood samples or with 200 µL of washed/unwashed blood from the experimentally infected goat #19 at 17 dpi (Cq-value 25.1) for two hours. Afterwards, the blood inoculum was removed, and flasks were refilled with medium supplemented with 10% FCS and antibiotics in double standard concentration (20,000 µg/mL Penicillin, 20,000 units/mL Streptomycin, 10 mg/mL Gentamicin, 250 µg/mL Amphotericin B). After 3 to 4 days of incubation at 37 °C, the infected BSR cell monolayer was split by using 1 mL of trypsin and mixed with 5 mL of the supernatant. In the next step, 3 mL of the cell–trypsin–supernatant suspension was transferred to a new T75 cm² cell flask with fresh BSR cells grown for 3 h. Three passages were performed, and the success of the virus replication was confirmed by the genomic load estimated by RT-qPCR. Furthermore, the virus presence was confirmed by the positive signal in the immune fluorescence test. Therefore, BSR cells were incubated for 4 h in 96-well cell culture plates and infected with the BTV-25 virus suspension. After 4 days of incubation at 37 °C and 5% CO2 a partial cytopathic effect (CpE) was visible and infected and non-infected BSR cells were fixated with 100 μL ice-cold Acetone-Methanol 1:1 for 10 min. After adding 100 µL of the 1:200 diluted BTV-25 rabbit immune serum, BSR cells were blocked with 100 μL ROTI^®^Block solution (Roth Chemie GmbH, Karlsruhe, Germany) for 30 min to reduce non-specific reaction. For the secondary antibody reaction, Goat anti-Rabbit IgG (Alexa Fluor^®^ 488, Abcam, UK) was prepared at a dilution of 1:1000 in ROTI^®^Block solution and 100 μL was added to each well. Fluorescence signaling was analyzed using an Axio Vert.A1 microscope (Zeiss, Oberkochen, Germany) with an HXP 120 V fluorescent light source.

### 2.7. ELISA

All serum samples were screened for BTV-group-specific antibodies using a cELISA (ID Screen^®^ Bluetongue Competition, ID-Vet, France) according to the manufacturer’s instructions. Samples with ≤50% of negativity compared to the negative control (S/N) were considered as positive, samples with ≥50% S/N as negative. 

### 2.8. Virus Neutralization Test 

A virus neutralization test was performed for the detection of serotype-specific neutralizing antibodies. BTV-25-GER2018 was used after 12 passages on BSR cells. In the last passage, the content of a T1700 cm^2^ cell roller flask was pelleted via centrifugation and re-suspended in 60 mL medium. For this VNT stock virus a titer of 10^5.83^ TCID_50_/_mL_ could be defined. VNTs were run with cELISA positive rabbit sera, sera of the experimentally infected goat, and all cELISA positive field samples of the goat flock. Furthermore, reference sera of classical BTV serotypes 1–24 (generated in guinea pigs or rabbits), and sera reactive against BTV-26, BTV-27v1 and BTV-28 were available for the VNT. A cELISA positive BTV-8 serum and a negative reference serum were used as positive and negative controls. Briefly, the serum was diluted in log2 steps starting from 1:10 to 1:280 and titrated against 100 TCID_50_ of BTV-25-GER2018 per 96 well. Plates were incubated for 1 h at 37 °C before overnight incubation at 4 °C. The following day, 100 µL of a BSR cell suspension of approximately 30,000 cells/100 µL was added per well. After incubation for 3–5 days at 37 °C, all wells were scored for a cytopathic effect (CpE). The neutralization titer was determined as the dilution of serum giving 100% neutralization. The calculations according to the Spearman and Kärber method were used.

## 3. Results

### 3.1. Genome Analysis

The sequences of all 10 segments of the BTV-25-GER2018 strain were established and used for phylogenetic analyses (Figure 1) including BTV strains representing the known BTV serotypes [6]. For segment 2, the nt identities for the BTV strains used in the phylogenetic tree varied from 40.9% (BTV-12) up to 60.8% (BTV-10) for the classical serotypes 1–24. The identity of the atypical serotypes started from 57.5% with BTV-28/Sheep pox vaccine derived BTV to up to 83.4% with TOV. For segment 6, identities for the classical serotypes from 58.0% (BTV-15) up to 72.2% (BTV-4 and BTV-24) were revealed. The identities for the atypical BTV serotypes varied from 68.5% for BTV-26 to the highest identity for TOV with 82.9%. In comparison, segment 10, a more conserved BTV segment, showed identities of 76.7% (BTV-18) up to 79.9% (BTV-21) with the classical BTV 1–24, and from 79.7% (SP vaccine derived BTV = BTV-28) up to 88.0% in comparison with atypical BTV (TOV).

The BLAST results of the nucleotide and amino acid sequences of the complete coding sequence of the BTV-25-GER2018 segments are shown in Table 1. The most related strains for all segments were found to be solitary representatives of atypical BTVs, which is consistent with the phylogenetic trees. The nearest neighbor (nt-based) for segment 1 was BTV-Z ITL2017 with 96.6%, and aa-based with 92.2% TOV. Segment 2 also matched with BTV-Z ITL2017 in nt and aa with 92.5% and 89.0% (query cover 79%), and as the second nearest neighbor with TOV in nt and aa with 83.5% and 82.8% (query cover 100%), respectively. The nearest neighbor of segment 3 was BTV-Z ITL2017 with 96.7% (nt) and the French BTV-27 variant 3 with 96.0% (aa). Segment 4 showed the highest identity in nt and aa with BTV-Z ITL2017 (96.1%/99.7%). The closest relatives for segment 5 were again BTV-Z ITL2017 (94.6%, nt level) and TOV (82.6%, aa level). Segment 6 matched to BTV-Z ITL2017 at 96.5% on the nt level (query cover 32%), whereas the second nearest neighbor was TOV reaching 82.9% (query cover 100%). Based on aa, segment 6 showed 91.8% identity with TOV. Segment 7 showed the highest identity with a Chinese BTV strain named “V196/XJ/2014”, both on the aa and the nt-level with 84.8% and 97.7%, respectively. For segment 8, the highest identities with 98.2% (nt) and 98.3% (aa) could be ascertained again with TOV. Similarly, for segment 9 the highest identity was found for TOV (85.3%/82.4%). For segment 10, the BTV-27/FRA/2014 variant 2 showed the highest identity on both the nt and aa level (88.8%/95.6%). 

BTV serotypes can be also divided into nucleotypes representing distinct evolutionary lineages [32]. In accordance with their serological cross-reactions and nucleotide identities, each BTV serotype clusters in a nucleotype group for segment 2 and segment 6, respectively. For the nucleotype classification of segment 2 their identity must be higher than 66.9% [33,34]. Currently 12 nucleotypes “A–L” are known for segment 2 with a putative 13th new nucleotype “M” involving BTV-28/Sheep pox vaccine derived BTV and a 14th nucleotype “N” involving BTV-Y TUN2017. Segment 2 of BTV-25-GER2018 belongs to nucleotype “K” together with TOV, BTV-X ITL2015, BTV-XJ1407 and the three variants of BTV-27. For segment 6, members of the same nucleotype need to show a >76% nt identity [34]. Currently, 10 nucleotypes “A–J” are known with the newest nucleotype “J” including BTV-27/FRA2014/v02, v03 and BTV-X ITL 2017 [6,35]. For segment 6, the newly described BTV-25-GER2018 strain is part of the nucleotype “H” together with TOV, BTV-27/FRA2014/v01, BTV-28/Sheep pox vaccine and BTV-XJ1407. Different BTV isolates are defined as serotypes by using the virus neutralization test, however, molecular typing is also possible. Within the same serotype, the minimum levels of Seg-2/VP2 sequence identities were defined as 68.4% nucleotide (nt) / 72.6% amino acid (aa) [24,34]. Segment 2 of BTV-25-GER2018 matched to 92.5% with BTV-Z ITL2017 as the closest neighbor, and with TOV to 83.5% as the second closest neighbor. Both are representatives of serotype 25. On the amino acid level, BTV-25-GER2018’s closest relatives are BTV-Z ITL2017 with 89.0%, and TOV with 82.8%. Overall, based on the sequence data analysis, it can be concluded that BTV-25-GER2018 belongs to the Seg-2 nucleotype grouping K for which TOV is the prototype isolate for BTV-25.

### 3.2. Goat Flock Monitoring

Results of the goat flock monitoring during the sampling period is shown in Table 2. Eleven goats were positive in the cELISA at all five bleeding time points, and four of those were continuously positive in the RT-qPCR as well, and one goat was continuously negative in the RT-qPCR. In total, 55 goats were negative in the cELISA at all five bleeding time points, five thereof were continuously positive in the RT-qPCR, and 33 were constantly negative. All data collected over the time are shown in detail in the Appendix A (Appendix A).

Figure 2 shows the median of all samples in the serogroup antibody-specific cELISA and the BTV-25 Seg-2 specific RT-qPCR analysis during the five bleeding time points. The median Cq-values developed from 35.96 to 34.48, 33.03, 33.58 and 35.72 at the fifth bleeding time point. For the cELISA, the median S/N% developed from 74 to 75, 77, 98 and 93, whereas regarding only the cELISA-positive results the S/N% values ranged from 25 to 24, 27, 27 and 14. 

Results of the RT-qPCR results and the cELISA assays for 20 new-born kids or goats newly added to the flock are shown in Table 3. The two individuals that arrived before December 2018 were negative in both the RT-qPCR and the cELISA at bleeding time points 3 and 4. Two of the 15 (13.3%) goats that arrived during spring 2020 were positive in the RT-qPCR for BTV-25 at bleeding 4, goat #127 with a positive cELISA result and goat #129 with a negative cELISA result. Interestingly, goat #127 was negative in both RT-qPCR and cELISA at bleeding time point 5, whereas goat #129 stayed positive in the RT-qPCR without seroconversion at bleeding 5. At this bleeding time point, 12 of 20 (60%) newly introduced goats were positive for BTV-25 viral RNA in RT-qPCR with no detection of group-specific antibodies. Only two of the goats—new at bleeding time point 5—were clearly positive in the cELISA, but negative in the RT-qPCR.

The analysis of the retrospective serum samples revealed the presence of BTV genomes in the goat flock since February 2015. In serum samples from three goats, evidence of BTV-25 nucleotype genomes could be detected in 2015 and viral RNA was present during the samplings in 2018/2019. In detail, the three goats originated from Switzerland, and all were present in the goat flock from February 2015 until the fifth bleeding in October 2019. Two goats were constantly positive for BTV-25 genome in RT-qPCR from 2015 until October 2019, one goat thereof was constantly negative in the cELISA, and one varied in the cELISA starting and ending with a negative result. The course of the RT-qPCR results of the third goat started with a positive test for the 2015 sample with variations over the different bleeding time points in 2018/2019 ending at bleeding 5 with a positive result again. The corresponding cELISA results of the third goat started with a negative result in 2015, followed by positive results during bleeding 1 to 3, and ending with a negative cELISA result again in 2019. The data are summarized in Appendix A. The BlueTYPE array runs were negative for other serotypes than BTV-25-GER2018.

### 3.3. Animal Experiments

Diagnostic inoculation of goats with two BTV-25-GER2018 RNA positive blood samples led to successful infection in one of the two animals. Viral RNA detection by RT-qPCR started on 7 dpi and peaked on day 17 with a Cq value of 25.1. The sera of the positive goat from 7, 14, 21 and 31 dpi increased in antibody titers measured by cELISA but did not reach the cut-off point to be classified as positive. The second experimentally infected goat remained BTV-genome negative throughout the animal trial and did not react with an antibody titer either. The respective data are shown in Table 4. Both goats did not show any clinical symptoms or fever at any time point during the trial. 

A polyclonal BTV-25-GER2018 antiserum was generated from the two immunized rabbits. The rabbit sera were clearly positive in the cELISA for group-specific BTV antibodies (log2 cELISA titers of up to 1:16). Nevertheless, no neutralizing titer could be determined for the rabbit sera due to an incomplete neutralization in the VNT assay.

### 3.4. BTV-25-GER2018 Isolation in Cell Culture

BTV-25-GER2018 was successfully isolated on BSR cells from washed blood samples of the experimentally infected goat #19 only. After the third cell culture passage, a CpE was observed and successful propagation of the virus could be confirmed with decreasing Cq-values in the RT-qPCR in higher cell passages. A maximum titer of 10^7.0^ TCID_50_ /mL was achieved after pelleting the cell culture material of passage 6, whereas the cleared supernatant fraction reached only 10^4.33^ TCID_50_/mL. The Pan-S10-RT-qPCR delivered a Cq value of 11.2 for the cell pellet and a Cq value of 16.3 for the cleared cell culture supernatant. 

### 3.5. Virus Neutralization 

A VNT using BTV-25-GER2018 was performed with the cELISA positive sera of the rabbit immunization trial, sera from the experimentally infected goats and all cELISA positive goat samples originating from the five bleeding time points. All these sera lead to the same result of an incomplete neutralization of BTV-25-GER2018. At only the 1:10 dilution step of the serum dilution, a CpE was observed in a part of the cell monolayer, and the CpE increased with the higher dilution steps until a 75–100% CpE was seen at dilution steps 1:80 or 1:160. The BTV-25-GER2018 negative reference serum showed a 100% CpE starting from dilution 1:10. Moreover, the reference sera of BTV serotypes 1–24 failed to neutralize BTV-25-GER2018, as well as the BTV-26, BTV-27x and BTV-28 specific sera. Figure 3 shows microscopy pictures of the partial virus neutralization effect of strain BTV-25-GER2018 in comparison to BTV-8, a representative of classical BTV. In contrast, the BTV-8-specific positive control serum showed the expected neutralization titer of 1:320 against the used BTV-8 strain.

## 4. Discussion

Here, we present the first isolation and characterization of a novel atypical BTV strain “BTV-25-GER2018”, detected in clinically healthy goats in a farm in Bavaria, southern Germany. Molecular analyses showed the highest identities on both the nucleotide and the amino acid sequence level with the group of atypical BTV-25 strains, which are clearly distinct from the canonical “classical” serotypes 1–24. Most importantly, BTV-25-GER2018 is the first BTV-25 strain, which could be efficiently propagated in cell culture. BTV-25-GER2018 did not cause any BTV typical clinical signs, neither in the goat flock nor in the experimentally inoculated goats. Infection with atypical BTV is characteristically not associated with clinical disease or only with very mild clinical signs [6,7,12,35,36]. BTV-25-GER2018 may have circulated in this flock for at least four years, as shown by retrospective analysis of serum samples. Serum is not recommended for BTV genome detection and only positive PCR results based on hemolytic sample materials can be accepted as true-positive. Furthermore, other serotypes than BTV-25-GER2015 present in the goat flock could have been excluded with the help of the BlueTYPE array [9] and are also unlikely from the epidemiological point of view. Germany was free of BTV from February 2012 until December 2018 and only <60 cattle tested positive since the BTV-8 re-emergence in the country [25,26]. No BTV-8 case in goats was reported in the area of sampling. 

A closer monitoring of BTV occurrence in the flock during five sampling time points in 2018/2019 revealed variations in the RT-qPCR and cELISA results from individual goats in the different consecutive samples. During the sampling period (from August 2018 until October 2019), 31% up to 38% of the goat flock tested positive for BTV genomes. Interestingly, all possible combinations of positive or negative RT-qPCR and cELISA results could be identified. Furthermore, the positive and negative results of individual goats varied with similar frequencies (Appendix A). A long-lasting RNA-positivity for up to 25 months of goats infected with TOV was observed previously in a goat flock in Switzerland [21]. Furthermore, TOV was found to be present in archived serum samples as early as 1998 [37]. Like in our study, no efficient viral clearance was observed in the Swiss goat flock. Possible re-infections with BTV, but above all the long-term viral persistence of BTV-25-GER2018 should be also considered as a possible scenario [21]. Unfortunately, no bleeding data between 2015 and 2018/19 were available for underlining the persistent infection model.

We observed a certain number of animals with only a weak antibody response, and goats with low BTV RNA levels during the monitoring period—both had also been reported for the Swiss BTV-25 related strain. These findings may be common for atypical BTV strains in contrast to the classical BTV 1–24 showing high RNA levels during the peak of viremia and long-lasting high-level antibody responses [21]. In the Bavarian goat flock, 18% to 23% of the goats were positive for group-specific antibodies in the cELISA during the monitoring period. In contrast to our findings, the BTV-25-infected Swiss goat flock showed a high in-herd seroprevalence of 97% in the cELISA [21]. In another seroprevalence study performed in Switzerland, the observed in-herd seroprevalence for goats was 75% in 2008 in the Swiss Alps (Valais), whereas the mean estimated in-herd prevalence in Ticino ranged from 33.5% to a maximum of 100% [37]. 

In the BTV-25-GER2018-positive goat flock, it seemed that goats had very variable genome loads (positive to weak positive to negative and positive again) independently of their antibody level. The “re-positivity” of several goats during the surveillance period could be interpreted as a re-infection, but more likely, the virus persisted in the goats without permanent viremia. Viral persistence mechanisms of BTV, in light of a “non-arthropod-based” overwintering, are an ongoing debate. One study suggested that BTV might persist in GammaDelta-T-cells [38], but this finding could not be reproduced, and it is currently not assumed that BTV could cause a persistent infection in ruminants [39,40]. Therefore, further research is needed to explore the potential for the long-term infection mechanisms of goats infected with atypical BTV strains including the novel BTV-25-GER2018.

The high variation in cELISA titers observed for the different sampling time points ranging even from seropositivity to seronegativity in several individuals has not been reported for BTV so far. In contrast, for classical BTV a long-lasting antibody response was observed in both sheep and cattle for at least 7.5 years [41,42]. In healthy individuals, true physiological fluctuations of antibody titers can result from polyclonal activation or depression of B-cells as observed for measles virus [43]. Long-term viral RNA persistence in cattle is known for Vesicular stomatitis virus, with fluctuations of IgG antibodies as shown in endemic areas such as Costa Rica. Reinfection of seropositive animals occurred as well [44]. The immune response of goats to atypical BTV strains needs to be further investigated to understand the observed antibody level variations. Nevertheless, a lower avidity of the VP7 (origin is a classical viral strain) used in the commercial cELISA towards the group-specific BTV antibodies produced to atypical viral strains cannot be fully excluded and may also contribute to a reduced sensitivity leading to false-negative results. The development of a cELISA using the VP7 of BTV-25 could illuminate the antibody reaction towards BTV-25-GER2018 in the goat flock. 

Neutralizing antibody levels are parameters of protective immunity towards the respective serotype [39,45]. For our studies no anti serato alternative atypical strains such as BTV-25 or BTV-Z-ITL2017, BTV-X ITL2015, BTV-Y TUN2017 and the Chinese BTV-XJ1407 isolate were available. Nevertheless, for sera specific for BTV-1 to 24, BTV-26, BTV-27 FRA2014-v01 and BTV-28, a VNT with BTV-25 GER2018 was performed and no cross-neutralization was observed. Interestingly, even the strong cELISA positive rabbit antiserum did not exhibit complete neutralization. In addition, all 125 ELISA-positive sera from the affected goat flock likewise did not show complete neutralization. It is rather unlikely that the other EDTA blood samples negative in cELISA show improved neutralization capacities. The lack of absolute neutralizing antibodies in those sera may contribute to the fact that BTV-25-GER2018 infections perpetuate for years in the goat flock without viral clearance and might be a part of the mechanisms supporting viral persistence of this atypical BTV. Nevertheless, prolonged viremia is known for BTV despite the presence of neutralizing antibodies due to its affinity to cell membranes and erythrocytes [39]. Unfortunately, as for the other BTV-25-related strains, virus isolation results have been unsuccessful so far [18,22], and only a recombinant BTV-25 chimera with a BTV-1 backbone (BTV-1^VP2/VP5 BTV25^) was available for neutralization assays as a positive control virus [20]. This recombinant BTV-25 virus could be neutralized by the TOV positive serum, which is in contrast to the here observed incomplete neutralization of BTV-25-GER2018 [18]. Nevertheless, the recombinant chimeric BTV-25 virus failed to be neutralized by the BTV-Z-ITL2017 positive serum [22]. For a better understanding of the pathogenesis and immune reactions of BTV-25-GER2018 and related strains, further research is necessary. 

The detailed analysis of samples from 15 newly introduced animals to the goat flock revealed that 13.3% of these were BTV-RNA positive in May 2019 at bleeding time point 4. In October 2019, at bleeding time point 5, the number of infected goats that had newly arrived increased to a level of 60%. The seasonality of competent vectors leads to the seasonality of BTV infections throughout the year with infection peaks during the late summer and autumn months [46] and might be the rationale for the increased infection rate in the newly introduced goats. In contrast, the number of BTV-25-GER2018 infections did not increase on a whole herd level in the autumn months. Therefore, alternative transmission ways and mechanisms of viral persistence relevant for BTV-25 related strains should be studied in the future.

Previously BTV-25 strains could not be propagated in cell culture until now [6,18,22]. For BTV-25-GER2018 virus propagation was successful on the mammalian cell line BSR, and we could show that BTV-25-GER2018 replication is cell-associated. This is in agreement with experiences from classical BTVs [47].

The genome sequence of BTV-25-GER2018 differed from other atypical BTV-25 strains. A small number of nucleotide exchanges within the BTV sequence detected in two genetically related strains could have a strong impact on virus characteristics, as demonstrated for two BTV-8 strains [48], and it is therefore very likely that differences on the molecular level have a major impact on virus isolation attempts. The difficulties of virus propagation in cell culture for BTV-25 and related strains, as well as the observed lack of neutralization for BTV-25-GER2018 and BTV-Z-ITL2015, lead to non-typeable phenotypes at least using traditional serotyping via VNT. For the atypical BTV strains, “genotyping” based on molecular data could be a practical solution. Finally, applying the criteria from Maan et al., 2016, BTV-25-GER2018 was genotyped as serotype 25. 

In conclusion, an atypical BTV, isolate BTV-25-GER2018, was found to circulate in southern Germany. It was identified as a member of the BTV-25 serotype group using sequence data and phylogeny. Retrospective samples confirmed the likely prolonged presence of BTV-25-GER2018 RNA within the studied goat flock. Furthermore, it was possible for the first time to propagate a BTV-25 related virus efficiently in cell culture. The analysis of consecutive samples from the affected goat flock suggests a persistent BTV-25-GER2018 infection in goats. This hypothesis is affirmed by the observation of mainly non-neutralizing antibodies against BTV-25 GER2018. Nevertheless, our findings are not conclusively proving the concept of persistent infection of goats with BTV-25 related viruses and further research on the biology of atypical BTV strains is necessary for a better understanding of their epidemiology and pathogenesis in comparison to the well-studied classical BTVs.

## Figures and Tables

**Figure 1 viruses-12-00983-f001:**
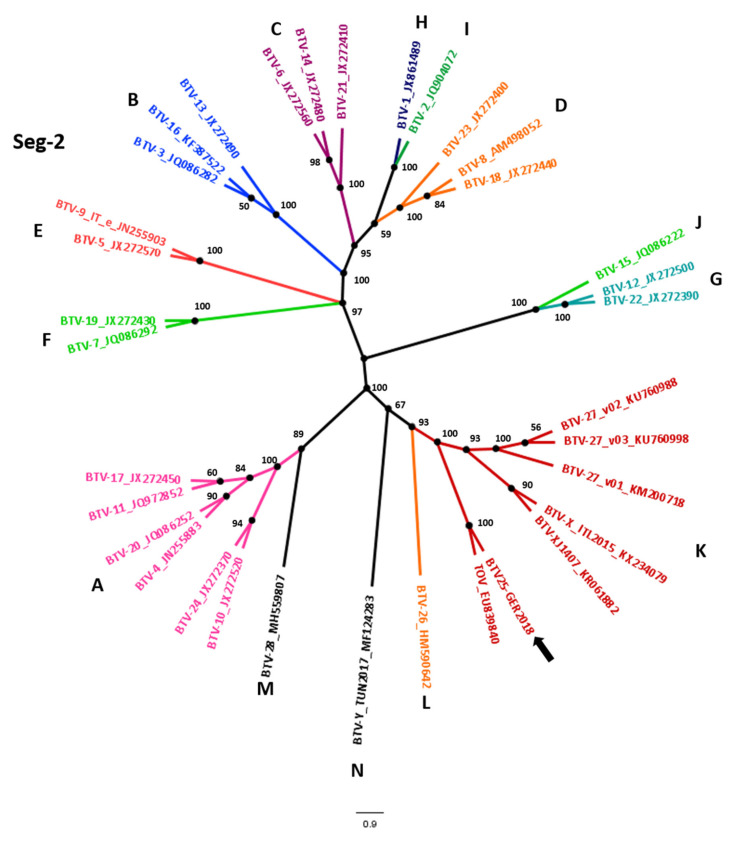
Phylogenetic analyses of the BTV-25-GER2018 genome. The phylogenetic trees of each of the 10 segments were created with MegaX using the genetic distinction model Tamura–Nei and tree-built method UPGMA including BTV strains representing the known BTV serotypes published [6]. We performed a bootstrap analysis with 1000 replications. The colors of the phylogenetic trees of segment 2 (Seg-2) and segment 6 (Seg-6) represent the different nucleotype groups [32] and trees were modified with FigTrees. For easier identification of the different nucleotypes A-N, the unrooted tree layout was chosen for Seg-2 and Seg-6. The arrows point at the BTV-25-GER2018 sequence.

**Figure 2 viruses-12-00983-f002:**
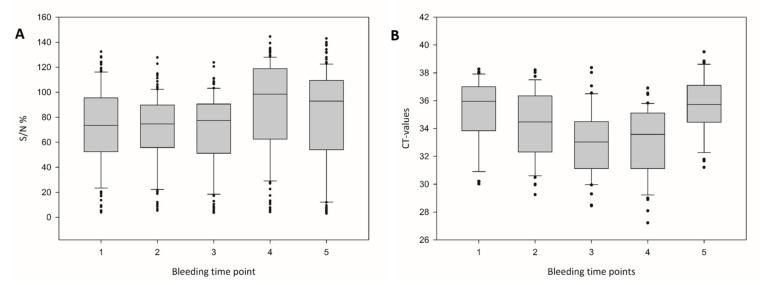
ELISA and PCR results of the 5 bleeding time points of the goat herd. The median values of the (**A**) reactivities of the ID.Vet cELISA in percent of negative control (≥50% is negative according to the manufacturer) and (**B**) the Cq values of the BTV-25 Mix13 Cq-values during the 5 bleeding time points of the goat flock are shown. The box and whisker plots show the median (broad central line), the interquartile range (box), the range of values (bars) and outliers (points).

**Figure 3 viruses-12-00983-f003:**
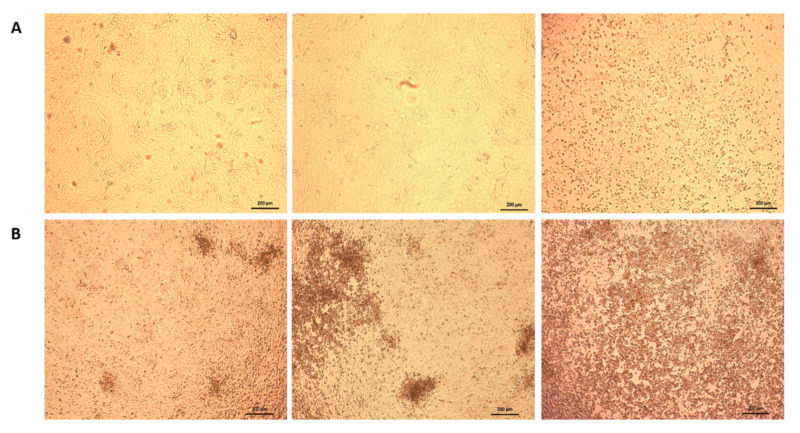
Comparison of BTV-8 and BTV-25-GER2018 virus neutralization. Virus neutralization results of (**A**) methodical control with BTV-8 virus and BTV-8 positive serum (1:10, 1:80 and 1:1280) (**B**) cELISA positive serum of a BTV-25-GER2018 field infected goat (1:10, 1:80 and 1:280).

**Table 1 viruses-12-00983-t001:** Closest neighbors of Bluetongue virus (BTV)-25GER2018 (CDS) following BLAST analyses. The three highest identities/similarities are presented.

	BLAST Best Hits				
Segment/ Protein (Accesion No.)	Serotype (nt/aa) ^#^	Strain (nt/aa)	Accession No. (nt/aa)	Identity Level % (nt/aa)	Query Cover %
1/VP1 (LR798441)	BTV-25/BTV-25	BTV-Z ITL2017/TOV	MF673720.1/ACY02806.1	96.55/92.17	88/100
	BTV-25/unknown	TOV/V196-XJ-2014	GQ982522.1/ASW41946.1	83.77/91.01	99/100
	BTV-27/BTV-27	BTV-27-FRA-2014-v02/BTV-27-FRA-2014-v01	KU760987/CEK41871.1	82.97/90.78	99/100
2/VP2 (LR798442)	BTV-25/BTV-25	BTV-Z ITL2017/BTV-Z ITL2017	MF673721.1/AVA16289.1	92.53/88.99	79/79
	BTV-25/BTV-25	TOV/TOV	EU839840.1/ACJ06702.1	83.48/82.79	100/100
	BTV-27/unknown	BTV-27-FRA-2014-v02/BTV-XJ1407	KU760988.1/AMM44543.1	74.54/75.18	100/100
3/VP3 (LR798443)	BTV-25/BTV-27	BTV-Z ITL2017/BTV-27-FRA2014-v03	MF673722.1/AMQ36829.1	96.74/96.00	63/100
	BTV-27/unknown	BTV-27-FRA-2014-v02/XJ1407	KU760989.1/AMM44545.1	85.18/95.89	100/100
	BTV-27/unknown	BTV-27-FRA-2014-v01/V196-XJ-2014	LN713672.1/ASW41948.1	85.18/95.89	100/100
4/VP4 (LR798444)	BTV-25/BTV-25	BTV-Z ITL2017/BTV-Z ITL2017	MF673723.2/AVA16291.2	96.13/99.66	90/91
	BTV-25/BTV-25	TOV/TOV	GQ982524.1/ACY02808.1	90.81/96.58	100/100
	Unknown/unknown	V196-XJ-2014/XJ1407	KX695173.1/AMM44546.1	80.88/93.79	100/100
5/NS1 (LR798445)	BTV-25/BTV-25	BTV-Z ITL2017/TOV	MF673724.1/ACJ06703.1	94.63/82.58	58/99
	BTV-25/BTV-28	TOV/ SPvvvv-02	EU839841.1/QGW56799.1	78.36/80.80	99/100
	BTV-8/BTV-28	CYP2016-04/BTV-28-1537-14	MN710167.1/QDH76488.1	75.12/80.62	99/100
6/VP5 (LR798446)	BTV-25/BTV-25	BTV-Z ITL2017/TOV	MF673725.1/ACJ06704.1	96.53/91.83	32/100
	BTV-25/BTV-27	TOV/BTV-27-FRA-2014-v01	EU839842.1/CEK41875.1	82.92/86.12	100/100
	BTV-27/BTV-28	BTV-27-FRA-2014-v01/BTV-28-1537-14	LN713675.1/QDH76491.1	78.24/82.70	100/100
7/VP7 (LR798447)	Unknown/unknown	V196-XJ-2014/V196-XJ-2014	KX695176.1/ASW41952.1	84.76/97.71	100/100
	Unknown/unknown	BTV-XJ1407/BTV-XJ1407	KR085416.1/AMM44548.1	83.90/97.42	100/100
	BTV-27/BTV-28	BTV-27-FRA-2014-v01/SPvvvv-02	LN713676.1/QGW56801.1	81.71/97.13	100/100
8/NS2 (LR798448)	BTV-25/BTV-25	TOV/TOV	EU839844.1/ACJ06706.1	98.21/98.30	100/100
	BTV-25/BTV-27	BTV-Z ITL2017/BTV-27-FRA-2014-v01	MF673726.1/CEK41877.1	97.28/86.69	58/100
	BTV-27/BTV-27	BTV-27-FRA-2014-v03/BTV-27-FRA-2014-v03	KU761004.1/AMQ36834.1	82.86/86.69	100/100
9/VP6 (LR798449)	BTV-25/BTV-25	TOV/TOV	EU839845.1/ACJ06707.1	85.35/82.37	100/100
	Unknown/unknown	BTV-XJ1407/V196-XJ-2014	KR085418.1/ASW41954.1	80.34/75.76	100/100
	Unknown/unknown	V196-XJ-2014/BTV-XJ1407	KX695178.1/AMM44550.1	79.16/75.68	100/100
10/NS3 (LR798450)	BTV-27/BTV-27	BTV-27-FRA-2014-v02/ BTV-27-FRA-2014-v02	KU760996.1/AMQ36826.1	88.84/95.63	100/100
	BTV-25/BTV-25	TOV/TOV	EU839846.1/ ACJ06708.1	87.97/94.32	100/100
	Unknown/unknown	BTV-X ITL2015/ BTV-X ITL2015	KX234087.2/ APC23697.2	82.87/93.89	99/100

^#^ (nt = nucleotide, aa = amino acide)

**Table 2 viruses-12-00983-t002:** Dynamics within the goat flock over the different sampling time points. The total numbers of positive goats in the BTV-25 specific RT-qPCR (quantification cycle (Cq) <40) and the cELISA (less than 50% negativity compared to the negative control (S/N)) are shown.

	Bleeding 1 07/08/2018	Bleeding 2 04/09/2018	Bleeding 3 03/12/2018	Bleeding 4 16/05/2019	Bleeding 5 08/10/2019
N° goats (EDTA/Serum)	118 (118/118)	117 (117/117)	116 (116/115)	121 (120/120)	115 (115/115)
New introduced	-	-	2	13	5
Removed	-	1	3	8	11
N° goats	118	117	115	120	115
PCR positiv	37 (31%)	45 (38%)	43 (37%)	44 (37%)	39 (34%)
cELISA positiv	27 (23%)	26 (22%)	27 (23%)	21 (18%)	24 (21%)

**Table 3 viruses-12-00983-t003:** BTV-25 specific RT-qPCR and cELISA results of the newly introduced individuals (offspring and newly introduced into the farm). Goats were considered positive for BTV-25 specific RT-qPCR with Cq <40 could be defined. The cut-off for the cELISA was 50% and serum samples with an S/N% ≤50% were determined as positive (in bold). “−“represents “no sample available” (not present at that time point in the goat flock).

	Bleeding 3 03/12/2018		Bleeding 4 16/05/2019		Bleeding 5 08/10/2019	
Goat ID	RT-qPCR	cELISA	RT-qPCR	cELISA	RT-qPCR	cELISA
	Cq-Value	S/N%	Cq-Value	S/N%	Cq-Value	S/N%
#119	no Cq	74	no Cq	115	32.8	93
#120	no Cq	106	no Cq	122	36.3	70
#121	−	−	no Cq	123	35.2	67
#122	−	−	no Cq	114	36.5	95
#123	−	−	no Cq	133	31.2	118
#124	−	−	no Cq	134	35.6	93
#125	−	−	no Cq	135	34.0	110
#126	−	−	no Cq	123	31.8	61
#127	−	−	32.1	28	no Cq	77
#128	−	−	no Cq	98	32.3	99
#129	−	−	30.8	84	31.7	102
#130	−	−	no Cq	132	34.4	70
#131	−	−	no Cq	98	no Cq	100
#132	−	−	no Cq	114	36.7	80
#133	−	−	no Cq	92	no Cq	79
#134	−	−	−	−	no Cq	68
#135	−	−	−	−	no Cq	92
#136	−	−	−	−	no Cq	107
#137	−	−	−	−	no Cq	16
#138	−	−	−	−	no Cq	20

**Table 4 viruses-12-00983-t004:** Results from experimental inoculation of goats. RT-qPCR (Cq-value) and cELISA (S/N%) results after diagnostic inoculation of two goats (#19 and #20) are shown.

Goat		dpi	0	3	5	7	10	11	12	14	17	18	21	24	31
#19	RT-qPCR		noCq	noCq	noCq	36.1	33.5	30.9	28.7	27.7	25.1	26.4	27.3	28.4	28.6
#19	cELISA					108				93			79		70
#20	RT-qPCR		noCq	noCq	noCq	noCq	noCq	noCq	noCq	noCq	noCq	noCq	noCq	noCq	noCq
#20	cELISA					101				107			106		108

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
