# Peer review of "Isolation and Cultivation of a New Isolate of BTV-25 and Presumptive Evidence for a Potential Persistent Infection in Healthy Goats"

_viruses, 2020, doi:10.3390/v12090983_

Round 1

Reviewer 1 Report

This manuscript describes the genomic characterisation and first time cell culture isolation of a novel German strain of BTV from within the 'atypical' BTV-25 nucleotype grouping (i.e. BTV-25-GER2018). The authors also provide circumstantial evidence, based upon a combination of qPCR and cELISA data, for long term persistence of this virus (perhaps up to 4.5 years) within the flock of goats from which it was isolated. The ability to fully characterise the genome of this new isolate and most importantly its successful isolation in cell culture make this a highly significant submission. I do however have some issues with the data used to make conclusions about long term persistence of this virus in the monitored goat flock. Specifically, I would like the authors to address the following points :

  • How can the authors assume the cELISA titres measured relate specifically to BTV-25-GER2018? The cELISA is serogroup (i.e. single VP7-epitope-specific) reactive, thus natural exposure of the herd to other BTV viruses (of any conventional serotype or atypical strain) would influence the result of a particular blood sample. For example, at least BTV-8 has been very active in Germany over the last few years. One way to counter this observation would have been the screening by the author's Seg-1 based qPCR assay (Ries et al. 2020) to at least distinguish whether any animals were infected by classical BTV strains during this period and relate this to the individual cELISA results from each animal? Could such data be easily generated and incorporated in this submission? In addition of course (and as conceded by the authors), the uniqueness of the VP7 aa sequence of BTV-25-GER2018 may also be severely compromising the ability of the cELISA to accurately indicate the true level of antibodies specific for BTV-25-GER2018 viral proteins that are present. The quoted cELISA titre of the rabbit antisera raised to the isolate (1/16) is also a very weak result. Perhaps the best approach would be to simply include an expanded range of animal bleeds in the VNT test (rather than utilise the cELISA results at all) using the rabbit antisera raised to the isolated virus. However the results gained with the antiserum itself and samples that were run in VNT do not seem that enlightening?
  • The data presented does not conclusively prove a persistent infection of the identified infecting strain. There is circumstantial evidence for quite prolonged periods of infection but the possibility of further separate incursions and infections of a second related strain of BTV being responsible for positive qPCR results cannot be discounted. In particular, for the period between 2015 and the bleeds from 2018 through 2019 their is no further bleed data presented. Thus the claim of persistence in three goats over a 4.5 year period can not really be convincingly substantiated.
  • Based upon the points made above, I would therefore strongly suggest the authors modify the Title of the submission accordingly. 'Isolation and cultivation (or propogation?)of a new isolate of BTV-25 and presumptive evidence for a potential for persistent infection in healthy goats' or something similar?
  • There are a couple of errors in the Supplementary Table. I think animal 53 should actually be listed in the 'Constantly Neg RT-qPCR and 'Thereof  variable in cELISA Section and animal 85 should be in the Thereof positive in cELISA Section of Positive RT-qPCR group?

If the authors modify their claims somewhat around evidence of viral persistence and respond adequately to the suggested grammatical changes and error corrections listed in the attached document, I would gladly recommend this submission for publication. The authors have certainly made a significant contribution and I hope to see the revised manuscript in print shortly

The manuscript is generally well written but I have suggested some  corrections that are listed in the attached document.

Author Response

Reviewer 1:

Comments and Suggestions for Authors

This manuscript describes the genomic characterisation and first time cell culture isolation of a novel German strain of BTV from within the 'atypical' BTV-25 nucleotype grouping (i.e. BTV-25-GER2018). The authors also provide circumstantial evidence, based upon a combination of qPCR and cELISA data, for long term persistence of this virus (perhaps up to 4.5 years) within the flock of goats from which it was isolated. The ability to fully characterise the genome of this new isolate and most importantly its successful isolation in cell culture make this a highly significant submission.

>Answer: We thank the reviewer for the benevolent evaluation of our work and the very helpful comments.

I do however have some issues with the data used to make conclusions about long term persistence of this virus in the monitored goat flock. Specifically, I would like the authors to address the following points:

  • How can the authors assume the cELISA titres measured relate specifically to BTV-25-GER2018? The cELISA is serogroup (i.e. single VP7-epitope-specific) reactive, thus natural exposure of the herd to other BTV viruses (of any conventional serotype or atypical strain) would influence the result of a particular blood sample. For example, at least BTV-8 has been very active in Germany over the last few years. One way to counter this observation would have been the screening by the author's Seg-1 based qPCR assay (Ries et al. 2020) to at least distinguish whether any animals were infected by classical BTV strains during this period and relate this to the individual cELISA results from each animal? Could such data be easily generated and incorporated in this submission?

>Answer: With the help of the BlueTYPE array including the suggested Seg-1 based qPCR assay of the reviewer (Ries et al. 2020) an infection with other serotypes could be excluded. Furthermore, the first BTV-8 case reported in Germany was in December 2018, when the BTV-25 occurrence was already ongoing for long. Between February 2012 and December 2018 Germany was officially free of BTV and in 2019 only a small amount of cattle (<60) was tested positive for BTV-8. In addition, BTV-8 in goats was never reported in general. The according information was included in the text:

#M &M, Line117: Furthermore, individual EDTA blood samples were tested in the BlueTYPE array as described previously [9].

#Results: Line 361: The BlueTYPE array runs were negative for other serotypes than BTV-25-GER2018.

#Discussion, Line 414: Furthermore, other serotypes than BTV-25-GER2015 present in the goat flock could have been excluded with the help of the BlueTYPE array [9] and are also unlikely from the epidemiological point of view. Germany was free of BTV from February 2012 until December 2018 and only <60 cattle were tested positive since the BTV-8 re-emergence in the country [23, 24]. No BTV-8 case in goats was reported in the area of sampling.

  • In addition of course (and as conceded by the authors), the uniqueness of the VP7 aa sequence of BTV-25-GER2018 may also be severely compromising the ability of the cELISA to accurately indicate the true level of antibodies specific for BTV-25-GER2018 viral proteins that are present. The quoted cELISA titre of the rabbit antisera raised to the isolate (1/16) is also a very weak result. Perhaps the best approach would be to simply include an expanded range of animal bleeds in the VNT test (rather than utilise the cELISA results at all) using the rabbit antisera raised to the isolated virus. However the results gained with the antiserum itself and samples that were run in VNT do not seem that enlightening?

>Answer: Indeed, the results with the VNT were not enlightening and therefore analyzing all serum samples via VNT not helpful. The neutralizing capacities of the rabbit sera and all other tested sera were not good, which means that also testing all samples would not enlight the results. The best approach would be the develepement of an ELISA using the VP7 of BTV-25, which can be target of the next study.

#Line 462 The development of a cELISA using the VP7 of BTV-25 could illuminate the antibody reaction towards BTV-25-GER2018 in the goat flock.

#Line 470: It is rather unlikely that the other EDTA blood samples negative in cELISA show an improved neutralization capacities.

  • The data presented does not conclusively prove a persistent infection of the identified infecting strain. There is circumstantial evidence for quite prolonged periods of infection but the possibility of further separate incursions and infections of a second related strain of BTV being responsible for positive qPCR results cannot be discounted. In particular, for the period between 2015 and the bleeds from 2018 through 2019 their is no further bleed data presented. Thus the claim of persistence in three goats over a 4.5 year period can not really be convincingly substantiated.

>Answer: The comment of the reviewer is correct, that the data are not conclusively proving the persistent infection. Nevertheless, combined with the observations made in former studies it is very likely. All minor comments of reviewer 1, qualifying the 4.5 year persistence were applied to the manuscript. Furthermore, the following additional explanations were made:

#Line 361 at the end of the result section goat flock monitoring: Removing of the following sentence: Overall, the circulation of BTV-25-GER2018 in the goat flock could be demonstrated for 4.5 years

#Line 427: Possible re-infections with BTV, but above all the long term viral persistence of BTV-25-GER2018 should be also considered as a possible scenario [21]

#Line 429: Unfortunately, no bleeding data between 2015 and 2018/19 were available for underlining the persistent infection model.

#Line 513: Our findings are not conclusively proving the concept of persistent infection of goats.

  • Based upon the points made above, I would therefore strongly suggest the authors modify the Title of the submission accordingly. 'Isolation and cultivation (or propogation?)of a new isolate of BTV-25 and presumptive evidence for a potential for persistent infection in healthy goats' or something similar?

>Answer: Title was changed according to the suggestion of the reviewer

  • There are a couple of errors in the Supplementary Table. I think animal 53 should actually be listed in the 'Constantly Neg RT-qPCR and 'Thereof  variable in cELISA Section and animal 85 should be in the Thereof positive in cELISA Section of Positive RT-qPCR group?

>Answer: Animals 53 and 85 are shifted into the correct sections

Minor comments:

If the authors modify their claims somewhat around evidence of viral persistence and respond adequately to the suggested grammatical changes and error corrections listed in the attached document, I would gladly recommend this submission for publication. The authors have certainly made a significant contribution and I hope to see the revised manuscript in print shortly

The manuscript is generally well written but I have suggested some  corrections that are listed in the attached document.

ABSTRACT

Line 12 remove 'such as'and replace with 'including'

# Such as was replaced with including

INTRODUCTION

Line 39 remove 'This bouquet'and replace with 'This group'

# Replaced with this group

Line 40 'differs' becomes 'differ'

# changed

Line 43 ideally need a suitable reference for the statement 'the OIE recommended'

# Reference added

Line 80 remove the hyphen between 'BTV' and 'situation'

# Hyphen removed

MATERIALS AND METHODS

Lines 122 and 128 'strain' should read 'strand' I think?

# Strain replaced by strand

RESULTS

Line 269 'represents the different nucleotype groups' should be followed by the Maan al. 2010 reference '[33]'.

# Reference added

Line 303 remove 'as mentioned before' this is unnecessary.

# As mentioned before is removed

Line 309 remove 'serotype 25'at the end of the sentence and replace with 'the Seg-2 nucleotype grouping K for which TOV is the prototype isolate for BTV-25' and add a reference to Fig 1. NB. ERROR IN FIG.1 as well -This nucleotype Group is mistakenly labelled as 'A' in the radial tree for Seg-2. Ideally too, I would prefer to see a better definition of East and West versions of BTV-9 isolates in both Seg-2 and Seg-6 radial trees.

# Correction of Line 309 was implemented and reference added. The nucleotype group label in Figure 1 was corrected and BTV-9 strains are labelled with BTV-9_IT_e (with e for eastern) and BTV-9_RSA_w (for western)

Line 311 remove 'The overview of the' and simply replace with 'Results of the goat....'

# Replaced

Lines 312 to 321 totally remove everything from 'At bleeding'right through to 'groupspecific antibodies'. All of the data stated in this text is clearly represented in Table 2. It is unnecessary to repeat it all in the text.

# The sentences were deleted

Line 321 between 'the' and 'cELISA' insert the term 'serogroup antibody -specific'

# Inserted

Line 322 between 'the' and 'RT-qPCR' insert 'BTV-25 Seg-2 specific'

# Inserted

Line 340 remove ‘The overview’ at the start of the sentence and replace whole line with ‘Results of RT-qPCR and CELISA assays for 20 new-born kids or goats newly’

# Replaced and changed

Line 356 remove ‘already’ insert ‘evidence of’ between ‘goats and BTV-25’ change ‘BTV25’ to ‘BTV-25 nucleotype’ Line 357 remove ‘already’ and remove ‘still’ insert ‘viral’ between ‘and’ and ‘RNA’

# All words inserted, ‘already and still’ are removed

Line 394 remove ‘Already in’ and replace with ‘At only’

# At only is written now in manuscript

DISCUSSION

Line 407 remove ‘discovery’ and replace with ‘first isolation’ replace ‘the novel; with ‘a novel’

# Removed and replaced

Line 411 insert ‘may have’ between ‘BTV-25-GER2018’ and ‘circulated’

# Inserted

Lines 411, 414 thru 418 remove ‘Interestingly and’ from the start of the sentence and start the sentence by capitalizing the ‘M’ of most. Then move this whole sentence to begin immediately after ‘1-24’ at the start of Line 411. Then next add the short paragraph within

Lines 416 thru 418 to the end of the previously moved sentence to form all of the first paragraph.

# Start of the sentence is adapted and the mentioned sentences shifted

Lines 411-414 Shift the two consecutive sentences starting with sentence ‘BTV-25GER2018…. And ending with the end of the following sentence ‘A closer……’consecutive samples.’ To the start of the second paragraph immediately before

# We changed the order in the best way according to the opinion of the reviewer, nevertheless due to the changes in relation with the major comments, the order of the sentence might have changed to the original proposal of the reviewer

Line 419 There should also be reference to the relevant Table in the manuscript for the statements made in these two sentences (i.e. Table S1).

# Reference to table S1 is added

Lines 422 and 423 remove the sentence starting with ‘Moreover’ and ending with ‘since February 2015’ you have ALREADY made this statement in the first paragraph of Discussion.

# Removed

Lines 425 and 426 remove the sentence starting with ‘Nevertheless and ending with ‘for up to 4.5 years’ As I have stated previously I don’t think you HAVE convincingly shown this.

# Removed

Lines 426 thru 431 Move the sentences starting with ‘A long-lasting …’and ending with ‘TOV[20].’To the end of the previously modified first paragraph and change the end of the last sentence of the moved block of text to ‘should also be considered as a possible(i.e delete ‘very realistic’) scenario.’ Delete rest of sentence after ‘scenario’.

# Moved to the first paragraph

Lines 433 and 434 insert ‘also’ between has and been. Remove ‘as well’ from the end of the sentence on Line 434. Remove ‘seem to’ and replace with ‘may’ on Line 434.

# Inserted and Removed according the suggestions

Line 447 remove ‘e.g.’

# Removed

Line 448 do not end sentence at ‘reproduced’ remove ‘However’ and replace with ‘and’

# Removed and sentence combined

Line 449 A second very relevant reference could be inserted here as well in addition to Reference [38]: Cultured skin fibroblast cells derived from bluetongue virus-inoculated sheep and field infected cattle are not a source of late and protracted recoverable virus R. A. Lunt1, L. Melville2, N. Hunt2, S. Davis2, C. L. Rootes1, K. M. Newberry1, L. I. Pritchard1, D. Middleton1, J. Bingham1, P. W. Daniels1, B. T. Eaton1 https://doi.org/10.1099/vir.0.81653-0 JGV 87(12)

# Reference is added

Also, on Line 449, replace ‘explain’ with ‘explore the potential for’

# Replaced

Line 464 insert ‘anti serato’ between ‘no’ and ‘alternative’ and remove related and replace with ‘atypical’. Add an ‘s’ to the end of ‘strain’ to make it plural.

# Inserted and replaced according to the reviewer

Line 465 change ‘was’ to ‘were’

# Changed

Line 491 Insert ‘Previously’ in front of BTV-25 to start the sentence

# Inserted

Line 505 insert ‘likely prolonged’ between ‘confirmed the’ and ‘presence’ and remove ‘for 4.5 years’.

# Inserted, the 4.5 years are removed

Reviewer 2 Report

Bluetongue virus serotype 25 causes long-term persistence in healthy goats

Corresponding author: B. Hoffmann

Since the description of the Toggenburg virus in Switzerland in 2008, several atypical Bluetongue virus (BTV) serotypes have been discovered. These serotypes share similar features as they infect small ruminants – mainly goats – with no clinical signs and an arthropod-independent transmission mode is suspected in most cases. The viraemia seems to persist for a longer period of time with these viruses as compared to classical BTV serotypes. Also in contrast to typical BTV strains, atypical viruses appear to trigger a weak and delayed immune response accompanied by inconsistent viral RNA detection.

In this article, Ries et al. describe for the first time the cell culture isolation of a BTV-25 related virus from a goat flock in Germany. They could show that this new virus was present since almost 5 years. They bring convincing data on the ability of this new BTV25 isolate to persist in goats. The information presented here is of great importance in our knowledge on the circulation, persistence, and other features regarding new BTV-atypical viruses. It will help to better understand transmission and pathogenesis traits of these viruses.

Author Response

Reviewer 2:

Comments and Suggestions for Authors

Bluetongue virus serotype 25 causes long-term persistence in healthy goats

Corresponding author: B. Hoffmann

 Since the description of the Toggenburg virus in Switzerland in 2008, several atypical Bluetongue virus (BTV) serotypes have been discovered. These serotypes share similar features as they infect small ruminants – mainly goats – with no clinical signs and an arthropod-independent transmission mode is suspected in most cases. The viraemia seems to persist for a longer period of time with these viruses as compared to classical BTV serotypes. Also in contrast to typical BTV strains, atypical viruses appear to trigger a weak and delayed immune response accompanied by inconsistent viral RNA detection.

In this article, Ries et al. describe for the first time the cell culture isolation of a BTV-25 related virus from a goat flock in Germany. They could show that this new virus was present since almost 5 years. They bring convincing data on the ability of this new BTV25 isolate to persist in goats. The information presented here is of great importance in our knowledge on the circulation, persistence, and other features regarding new BTV-atypical viruses. It will help to better understand transmission and pathogenesis traits of these viruses.

>Answer: We thank the reviewer for the benevolent evaluation of our work.